# Targeting Casein Kinase 1 (CK1) in Hematological Cancers

**DOI:** 10.3390/ijms21239026

**Published:** 2020-11-27

**Authors:** Pavlína Janovská, Emmanuel Normant, Hari Miskin, Vítězslav Bryja

**Affiliations:** 1Department of Experimental Biology, Faculty of Science, Masaryk University, 62500 Brno, Czech Republic; janovska@sci.muni.cz; 2TG Therapeutics, New York, NY 10014, USA; enormant@tgtxinc.com (E.N.); hm@tgtxinc.com (H.M.); 3Department of Cytokinetics, Institute of Biophysics, Academy of Sciences of the Czech Republic, 61265 Brno, Czech Republic

**Keywords:** casein kinase 1, CK1α, CK1ε, leukemia, CLL, AML, MM, inhibitors, umbralisib, WNT pathway

## Abstract

The casein kinase 1 enzymes (CK1) form a family of serine/threonine kinases with seven CK1 isoforms identified in humans. The most important substrates of CK1 kinases are proteins that act in the regulatory nodes essential for tumorigenesis of hematological malignancies. Among those, the most important are the functions of CK1s in the regulation of Wnt pathways, cell proliferation, apoptosis and autophagy. In this review we summarize the recent developments in the understanding of biology and therapeutic potential of the inhibition of CK1 isoforms in the pathogenesis of chronic lymphocytic leukemia (CLL), other non-Hodgkin lymphomas (NHL), myelodysplastic syndrome (MDS), acute myeloid leukemia (AML) and multiple myeloma (MM). CK1δ/ε inhibitors block CLL development in preclinical models via inhibition of WNT-5A/ROR1-driven non-canonical Wnt pathway. While no selective CK1 inhibitors have reached clinical stage to date, one dual PI3Kδ and CK1ε inhibitor, umbralisib, is currently in clinical trials for CLL and NHL patients. In MDS, AML and MM, inhibition of CK1α, acting via activation of p53 pathway, showed promising preclinical activities and the first CK1α inhibitor has now entered the clinical trials.

## 1. Introduction

Phosphorylation, mediated by protein kinases, is a common posttranslational modification with 500,000 potential phosphorylation sites in the human proteome and 25,000 phosphorylation events described for 7000 human proteins [1]. Since protein kinases play dominant roles in the regulation of a wide range of cellular functions including initiation of cancer, tumor progression and the development of metastatic diseases, many of them represent attractive targets for modern medicine. Some of these kinases have been targeted by novel therapeutics (protein kinase inhibitors) for treatment of cancer [2]. A vast array of kinase inhibitors are under clinical investigation and 52 small molecule kinase inhibitors have been already approved by the U.S. Food and Drug Administration (FDA) [3].

Whereas some protein kinases have been long recognized as attractive targets, others have come to the forefront of biomedical research only recently. These include the family of casein kinase 1 (CK1). CK1 proteins are important regulators of signal transduction, involved in diverse signaling pathways and acting in most cell types. In human, the CK1 protein family includes seven isoforms, α, γ1, γ2, γ3, δ, ε and less well described α-like, with CK1α, δ and ε being the most studied to date [4]. All CK1 proteins are serine/threonine, monomeric protein kinases. All CK1 enzymes share high homology in their kinase domains (53–98%) [5]. However, each isoform has a different length of N-terminal (8–45 amino acids (aa)) and C-terminal (51–138 aa) regions. The CK1δ and ε isoforms share the highest homology in the primary structure and have a long and similar regulatory C-terminus [6]. The crystal structure was solved for all human CK1 isoforms alone or in complex with specific inhibitors: CK1α-Protein Data Bank (PDB) structure 5FQD [7], 6GZD [8], CK1γ1-3-2CMW, 2C47, 2IZU, CK1δ-3UYT [9], CK1ε-4HNI [10] (Figure 1).

## 2. Biological Functions of the CK1 Family

CK1 family members are involved in various cellular processes such as mitotic checkpoint signaling, DNA repair, apoptosis and p53 pathway, protein translation, circadian rhythm, endocytosis and autophagy, immune response and inflammation, centrosome-associated processes, and play a key regulatory role in several developmental pathways such as Wnt, Hedgehog, NF-κB or Yap/Taz signaling (for reviews see [4,12,13,14]). The CK1 kinases commonly act as both positive and negative regulators of these processes, with redundant, distinct or opposite roles of individual CK1 isoforms. Dysregulation of widespread regulatory network controlled by CK1 kinases may result in various pathological conditions, including cancer development, metastasis spreading or neurodegenerative diseases [12,15]. Most of these aspects of CK1 biology are well covered in the abovementioned recent review articles. Herein, we will thus briefly summarize CK1 functions only in those biological processes that are relevant for the topic of this review, i.e., the role of CK1 in hematological malignancies.

CK1ε and redundantly functioning CK1δ are well-established positive regulators of Wnt signaling pathways [16,17,18,19,20]. The Wnt signaling pathways can be classified as either Wnt/β-catenin (canonical) or non-canonical. When a specific WNT ligand (there are 19 *WNT* genes) binds to a specific Frizzled (FZD) receptor (there are 10 *FZD* genes [21]), a co-receptor associates with the FZD receptor. According to the current understanding, if the co-receptor is the low-density lipoprotein receptor related protein 5/6 (LRP5/6), the newly formed hetero-trimer WNT-FZD-LRP recruits a group of proteins described as the “signalosome” that activates the Wnt/β-catenin pathway, in what is described as the canonical pathway (Figure 2). If the co-receptor, however, is receptor tyrosine kinase-like orphan receptor (ROR) 1 or ROR2 or RYK, the newly formed trimer WNT-FZD-ROR (or RYK) recruits another set of protein that leads to the activation of the Wnt/Ca^2+^ pathway or the Wnt/planar cell polarity (PCP) pathway, in what is called the non-canonical pathway. The same Frizzled can bind either a ligand that preferentially activates the canonical (WNT-2, 3, 3A and 8) or the non-canonical (WNT-4, 5A, 5B, 6, 7a and 11) branch of the Wnt signaling [22].

Dishevelled (DVL) is a phosphoprotein and a crucial CK1δ/ε substrate that is required for both canonical Wnt/β-catenin and non-canonical Wnt signaling. Wnt activation stimulates CK1δ/ε kinase activity, promoting phosphorylation of DVL that results in the signal propagation and downstream effects [16,17,33]. In the canonical Wnt pathway it includes the stabilization of β-catenin, which then enters the nucleus and activates the transcription of genes which regulate cell proliferation and differentiation [34,35]. In the non-canonical Wnt pathways, different downstream effects can take place such as activation of Ca^2+^ release but also, for example, phosphorylation of NFAT [36] or NF-κB [37]. Although CK1α, CK1γ, and CK1ε have all been implicated in Wnt signaling pathways, only CK1ε and closely related CK1δ have been shown to be required for the WNT-induced phosphorylation and activation of DVL [17]. As such, CK1δ/ε represents an attractive therapeutic target that impacts both canonical and non-canonical Wnt pathways.

Of note, all members of CK1 kinase groups (α, γ and δ/ε) have been described to have distinct and non-redundant functions in vivo [4]. For example, in the Wnt/β-catenin pathway, CK1α and CK1δ/ε appear to have opposite roles. CK1α acts as a component of the β-catenin destruction complex where it phosphorylates β-catenin to promote its degradation, whereas CK1δ/ε mediate upstream WNT signal by phosphorylating DVL and promoting stabilization of β-catenin. As a consequence, CK1δ/ε inhibition blocks Wnt/β-catenin pathway, in contrast to CK1α inhibition that activates it. In general, isoform specific biology of individual CK1 family members is poorly defined and it is not always clear when—or if—CK1 isoforms have redundant functions and when a biochemical event is controlled uniquely by a particular CK1 isoform(s). The experiments using CK1 inhibitors therefore, in this respect, should be interpreted carefully.

## 3. CK1 Inhibitors and Their Therapeutic Potential in Hematologic Malignancies

Because of the abovementioned involvement of different CK1 isoforms in key biological processes, the CK1 kinases have been connected to the pathogenesis of diverse disorders, including cancer [4,12]. In particular, Wnt pathways controlled by CK1 were shown to be essential in driving hematologic malignancies [38,39,40]. Importantly, individual CK1 isoforms can act as tumor suppressors as well as oncogenes, depending on the disease, tissue, cell type and cellular process in question [4,12].

Pharmacological inhibition of the CK1 family members has been investigated as potential therapy in preclinical studies involving neurodegenerative diseases [41], obesity [42], behavioral disorders [43], alcohol [44] or opioid addiction [45] and several cancer types [12]. Regarding the anti-tumor efficacy of CK1 inhibition, for example, it has been shown that CK1δ/ε inhibitors with low nanomolar IC_50_s demonstrated efficacy in vitro and in preclinical in vivo models of breast cancer (e.g., SR-3029 [46]). Some CK1 inhibitors proved to have higher efficacy on mutant CK1 variants (e.g., CK1δ mutants in colon cancer [47]), which may be beneficial as a form of personalized medicine. These preclinical studies have involved both in vitro and in vivo stages of testing, and a number of structurally different compounds. For a recent comprehensive summary of CK1 inhibitors please see Knippschild et al. [4].

Currently, hematological malignancies represent the only therapeutic area of cancer research where kinase inhibitors targeting the CK1 family have reached the clinical stage (see below). To orient the reader, we have summarized (Table 1) the properties of CK1 inhibitors that were used in the initial studies that established the potential of CK1 inhibition in lymphoma and leukemias described later in Section 4 and Section 5

The first described CK1 inhibitors CKI-7 (IC_50_ = 6 μM for CK1δ) [48] or IC261 (IC_50_ = 1.0 μM for CK1 δ/ε, IC_50_ = 16 μM for CK1α) [49] were not very specific but showed activity against pancreatic tumors in a xenograft mouse model [50] as well as in a MYC-driven neuroblastoma model [51]. However, data for IC261 should be interpreted carefully as this compound also inhibits mitotic spindle formation via a mechanism that does not involve CK1 inhibition [52]. D4476, reported as a CK1δ and ALK5 inhibitor (IC_50_ = 200 nM and 500 nM, respectively) [48] has been shown to also block CK1α both in vitro [53] and in cells [17], and has therefore been widely used to describe CK1α biology. 

The newer and more selective generation of inhibitors—PF-670462 and PF-4800567—have helped to isolate and distinguish the biological effects of CK1δ and ε, mainly in the studies on circadian rhythm [54,55]. To date, PF-4800567 is unique with its 22-fold greater potency toward CK1ε than CK1δ (IC_50_ = 32 nM and 711 nM respectively) [54]. The list of inhibitors described in Table 1 concludes with two compounds in clinical trials, umbralisib and BTX-A51, that will be discussed in greater detail below. For summary of the selectivity profile and the most common off-targets of the discussed compounds, see Table 1.

Overall, no isoform-specific CK1α or CK1γ inhibitors have been produced so far [58]. The potential of CK1γ inhibitors in the treatment of hematologic malignancies is not clear due to the limited overall data available regarding its biology. Recent report, however, shows that CK1γ2 may be implicated in chronic lymphocytic leukemia (CLL) and other B cell lymphomas as one of the targets of microRNA-155 [59]. The potential of CK1δ/ε-specific and CK1α-specific inhibitors in leukemia, lymphoma and myeloma therapy is better described, and will be discussed below, including the rationale for targeting CK1δ/ε in B cell lymphomas and CK1α in myelodysplastic syndrome (MDS) and acute myeloid leukemia (AML).

## 4. CK1ε in Chronic Lymphocytic Leukemia (CLL) and Other Non-Hodgkin Lymphomas (NHLs)

CLL is a lymphoproliferative disease characterized by clonal expansion of mature antigen-experienced CD5+ B cells in the peripheral blood (PB), lymphoid tissue and bone marrow (BM) [60]. It is the most common adult leukemia in Western countries; with a median age of diagnosis 67–72 years [60,61]. The disease course is highly heterogeneous—while some CLL patients remain asymptomatic, others develop an active disease with one or more symptoms requiring therapy. CLL therapy typically involves immunochemotherapy and, more recently, targeted inhibitors blocking pro-survival B cell receptor signaling (BCR, e.g., ibrutinib, idelalisib) or anti-apoptotic BCL2 signaling (venetoclax) [60]. Despite the significant improvement in patient response to the therapies tailored for specific risk-groups, these targeted therapies need to be mostly infinite to prevent relapse. Novel strategies currently tested in clinical trials involve drug combinations or sequential treatments by drugs targeting different signaling pathways in an effort to increase depth of response and define time limited therapeutic regimens.

### 4.1. CLL and the Non-Canonical Wnt Pathways

The multiple links between Wnt signaling pathways and B-cell leukemia and lymphoma have been reviewed recently [40]. Here, we will thus highlight only the key findings relevant to the topic of this review.

Many non-canonical Wnt pathway components, including CK1δ/ε and their substrates DVL proteins, are overexpressed in CLL [28,32,62,63]. The most striking upregulation, however, is the expression of Receptor Tyrosine Kinase-Like Orphan Receptor-1 (ROR1)—a membrane receptor for WNT-5A— that is uniquely upregulated in CLL cells [29,64,65] and in most other mature-B cell lymphomas [66], while its expression in normal B cell or leukocyte populations is very limited. A similar expression pattern of ROR1 and other non-canonical Wnt pathway components is observed in mantle cell lymphoma (MCL) [67]. MCL is a specific type of non-Hodgkin B cell lymphoma bearing the recurrent t(11;14)(q13;q32) chromosome translocation which brings the cyclin D1 gene under the influence of the enhancer of the immunoglobulin heavy chain (IgH) gene, leading to cyclin D1 overexpression [68]. The expression pattern of ROR1 has made it a preferred target for the development of specific anti-cancer therapeutics including monoclonal antibodies (cirmtuzumab [69]), CAR-T cells [70] or small molecule inhibitors [71,72,73,74] which block the phosphoinositide 3-kinase (PI3K)/AKT axis and exert similar biological effects as PI3K/AKT inhibitors [75]. Of note, CK1ε was shown to bind ROR1 and act downstream to regulate the PI3K/AKT axis [76,77], similarly to ROR2 [78]. CLL pathobiology thus seems to be closely connected with non-canonical Wnt pathway and CK1δ/ε, which are known to be crucial regulators of this pathway.

The ROR1-driven non-canonical Wnt pathway is involved in the communication of CLL cells with their microenvironment [30,79]. The WNT-5A/ROR1 axis has been shown to regulate CLL cell motility, chemotaxis, survival and proliferation [28,62,80,81,82,83,84,85]. All these processes have been linked directly to CLL/B cell lymphoma pathogenesis and have been shown to act in a similar fashion in MCL [86] or Hodgkin lymphoma [87]. Overall, high expression of non-canonical Wnt pathway genes and proteins was shown to correspond with poor outcome of CLL patients [28,62,88], deregulated motility (specifically for CLL cells with high autocrine expression of the Wnt5a ligand) and chemotaxis towards CXCL12 and CCL19 chemokines [28,30,89].

### 4.2. Inhibition of CK1 δ/ε in the Preclinical Models of CLL

In addition to gene expression and in vitro studies, the key role of the non-canonical Wnt pathway in CLL pathogenesis was confirmed by independent genetic experiments in the Eμ-TCL1 mouse model of CLL [89,90]: Overexpression of ROR1 in B cells promoted disease development [89] whereas the loss of FZD6, another receptor of the non-canonical Wnt pathway highly expressed in the leukemic B cells in this transgenic model of CLL, caused significant delay in the disease development [90].

CK1ε and CK1δ act downstream of these receptors and represent a possible therapeutic approach for CLL and other B cell lymphomas. Indeed multiple CK1 inhibitors—D4476, PF-4800567 and PF-670462—were able to block efficiently CXCL12 and CCL19-induced chemotaxis of CLL cells [28,30] and D4476 effectively blocked infiltration of lymphoid organs by CLL cells in the patient-derived xenograft model based on NOD/SCID IL2rγnull mice [28]. PF-670462, well characterized for its high potency and safety in vivo [54], blocked polarity acquisition in CLL cells in vitro [91] and prolonged survival of Eμ-TCL1 mice that spontaneously develop CLL [30]. In the same study CK1δ/ε inhibition also blocked the migration of CLL cells derived from all—including the high-risk—patient subgroups. Furthermore, CK1 inhibition blocked the functional response of CLL cells in co-culture with the bone marrow stromal cells that was demonstrated by the decreased production of CCL3/4 [30], two chemokines essential for the CLL-T cell interaction in the lymphoid tissue microenvironment [92].

### 4.3. Umbralisib, A Dual PI3Kδ-CK1ε Inhibitor, Demonstrates Safety of CK1ε Inhibition

In the context of CLL, it is important that an orally bioavailable phosphoinositide-3-kinase delta (PI3Kδ) inhibitor, umbralisib (also known as TGR-1202), currently being studied in several hematologic cancer indications, was identified to also act, at higher concentrations, as a CK1ε inhibitor [57]. It is currently the only known compound able to block CK1ε kinase activity that has undergone all toxicology and safety phases of clinical testing in human (recently completed phase III). Plasma levels of umbralisib were found to exceed 5 μM at steady state [93], which well exceeds the concentration shown to inhibit CK1ε in cells [57]. As such, analysis of the effects of umbralisib provides a unique opportunity to assess the clinical potential of CK1ε inhibitors.

A FDA filing for umbralisib for patients with marginal zone lymphoma (MZL) and follicular lymphoma (FL) was recently accepted for review [93,94]. In clinic, incidences of immune-mediated toxicities have consistently been lower with umbralisib than with the approved PI3Kδ inhibitors idelalisib and duvelisib which do not inhibit CK1ε (Table 2), the potential mechanism for which may be related to CK1ε inhibition, as described below [93,95].

Two recent studies have assessed the possible contribution of CK1ε inhibition to the clinical effects of umbralisib by comparing umbralisib to duvelisib and/or idelalisib, two approved PI3K inhibitors devoid of CK1 inhibitory activity [57,96]. In the first study, authors asked why umbralisib and not idelalisib showed specific cytotoxicity in CLL, lymphoma and myeloma cells when combined with proteasome inhibitor carfilzomib [57]. Umbralisib was capable of blocking phosphorylation of eukaryotic translation initiation factor 4E binding protein (4E-BP1), which resulted in the inhibition of c-MYC translation and subsequent cell death. Idelalisib, unable to trigger this mechanism on its own, reduced MYC levels only when combined with CK1ε-specific inhibitor PF-4800567. This suggested that inhibition of CK1ε is essential for these effects—in line with an earlier report demonstrating regulation of 4E-BP1 phosphorylation and messenger RNA (mRNA) translation by CK1ε [97]. These results suggest that umbralisib’s unique ability to inhibit CK1ε may increase its therapeutic window by increasing anti-tumor activity [57].

In the second study [96], authors analyzed the reasons for lower immune-mediated toxicity of umbralisib in comparison with other FDA-approved PI3K inhibitors. They hypothesized that CK1ε inhibition might lead to an increase of regulatory T-cell (Treg) activity, immune suppression, and thereby lower the risk of immune-mediated toxicities frequently observed with prior PI3K inhibitors. In an in vivo experiment, Eμ-TCL1 mice treated with idelalisib and duvelisib had higher intestinal and liver inflammation grades compared to umbralisib. A reduced number of peripheral Tregs observed with idelalisib and duvelisib positively correlated with overall inflammation grade in Eμ-TCL1 treated mice. A CK1ε specific inhibitor, PF-4800567, significantly increased Treg counts when added to all three PI3K inhibitors, suggesting that inhibition of CK1ε by umbralisib contributed to its Treg sparing effects. The molecular mechanism is unclear but can involve the inhibition of Wnt/β-catenin pathway by CK1ε inhibition since it is known that activation of the canonical Wnt pathway triggers T cell receptor transcription factor TCF-1-dependent induction of IL2, followed by the suppression of FoxP3 transcriptional activity and decreased Treg function [98].

In summary, umbralisib is a very selective dual inhibitor. Despite relatively lower potency as a CK1ε inhibitor, clinical data from umbralisib show that CK1ε inhibition is well tolerated and brings therapeutic benefit in CLL, opening the possibility that a more potent and selective CK1δ, CK1ε or CK1δ/ε inhibitor could demonstrate activity especially in combination with other compounds and become an important drug for the treatment of hematological cancers.

### 4.4. Potential of the Combinations of the B-Cell Receptor (BCR) and CK1 Inhibition in CLL

The majority of cancer treatments today are comprised of combinations of multiple drugs with distinct mechanisms of action. In CLL and MCL, for example, both, BCR and non-canonical Wnt pathway mediated by WNT-5A/ROR1, represent intensively exploited therapeutic targets [60,67,69]. BCR inhibition achieved by Bruton tyrosine kinase (BTK) inhibitor ibrutinib represents effective therapy for many CLL patients but does not induce complete responses without continued therapy [60]. This suggests that non-BCR pathways, such as non-canonical Wnt pathway, also contribute to CLL growth. The combinatorial treatment of BCR and non-canonical Wnt signaling can be of particular importance in CLL and MCL where it has been shown earlier that in these malignancies the non-canonical Wnt pathway and BCR signaling can be targeted in a combinatorial manner. Namely, it has been observed in vivo in the mouse models of CLL where BTK inhibitor ibrutinib and anti-ROR1 antibody cirmtuzumab showed synergistic effects [99]. Importantly, similar synergism has been described in the same mouse model for ibrutinib and CK1 inhibitor PF-670462 [30]. Similar behavior has been observed by Karvonen and colleagues in the in vitro model of MCL [67]. This suggests that Wnt pathway inhibitors, including inhibitors of CK1(δ)/ε, can synergize with BTK inhibitors like ibrutinib in patients with CLL or other ROR1^+^ B-cell malignancies such as MCL.

Interestingly, umbralisib, the PI3Kδ and CK1ε dual inhibitor discussed in the previous section, has recently been tested in CLL and MCL relapsed or refractory patients, in combination with ibrutinib in presence [100] or absence [95] of a novel glycoengineered anti-CD20 antibody, ublituximab. In the triple combination study, treatment was well tolerated with a manageable adverse event profile, and efficacy data was available from 44 patients demonstrating the triple combination to be highly active [100]. The combination ibrutinib-umbralisib was also very effective in a relapse or refractory population of CLL and MCL patients. No dose-limiting toxicities were observed and the maximum-tolerated dose of umbralisib was not reached. The overall response was 90 % in CLL and 67 % in MCL [95].

## 5. CK1α in Myelodysplastic Syndrome (MDS) and Acute Myeloid Leukemia (AML)

The therapeutic potential for CK1 inhibitors in hematologic malignancies is not limited to CLL and MCL. Another interesting clinical potential for CK1 inhibition comes from studies on the role of CK1α in MDS and AML. These diseases vary greatly from CLL or MCL that originate from mature B cells.

MDS represents a diverse spectrum of disorders characterized by variable degrees of refractory cytopenia and morphological dysplasia of precursor and mature BM blood cells [101]. It is a group of disorders with a common origin in hematopoietic stem cells (HSCs), but heterogeneous biological and genetic characteristics and clinical manifestations, with incidence ~4 cases per 100 000/year and median diagnosis age 70–75 years. Therapeutic options for MDS patients vary depending on the disease course and present risk factors and range from supportive care to allogeneic stem cell transplantation.

MDS can transform to AML in up to 30% cases [101]. AML, similar to MDS, is characterized by increased proliferation and impaired differentiation of hematopoietic stem and progenitor cells. It includes a variety of disorders defined by accumulation of abnormal myeloblast in BM and frequent involvement of PB, leading to BM failure [102,103] and a cut-off of 20–29% of blasts present in BM is used to distinguish these two entities [101]. AML incidence (~5 cases per 100 000/year in Europe) is age dependent and rising in patients over 60 years, with median age of diagnosis ~70 years [103].

MDS and AML share a number of common features, including a set of recurrent cytogenetic aberrations, gene fusions and mutations [104] that have strong prognostic significance, and guide patients risk-stratification and choice of therapy [101,105]. It is, however, clear that despite the progress brought by deeper understanding of AML genetic background and the associated disease biology, the 5-year overall survival in adult AML remains <20% [102,103,106] and therefore the development of novel therapies with higher response rates, greater efficacy and lower toxicity are crucial goals of the ongoing research.

### 5.1. CK1α Activity is Critical for AML Leukemic Stem Cells (LSCs) Survival and Growth In Vivo

CK1α encoding gene (*CSNK1A1*) is located in the region commonly deleted in MDS and AML–del(5q32) [107,108]. Chromosome 5q deletions belong to the most common cytogenetic aberrations in MDS and the subgroup of patients which carry isolated del(5q) have a distinct clinical phenotype [109,110,111]. Genes present in this region do not undergo homozygous inactivation but are typically in the state of haploinsufficiency [112].

The effects of CK1α haploinsufficiency on adult hematopoiesis has been studied by Schneider et al. [107] using a conditional knockout mouse model. Homozygous *Csnk1a* knockout in HSC resulted in rapid lethality (5–17 days) due to BM failure accompanied by the increased number of cells exiting quiescence and entering cell cycle, followed by apoptosis induction. *Csnk1a* haploinsufficient cells, however, recovered and managed to outgrow the control cells despite initially slower repopulation in secondary/tertiary transplants. Importantly, *Csnk1a* haploinsufficient cells were more sensitive to a small-molecule CK1 inhibitor D4476 [48] that caused selective depletion of HSCs and progenitor *Csnk1a* deficient cells in the BM.

Additional evidence came from the short hairpin RNA (shRNA) screen in a MLL-AF9 mouse model of AML [113] that identified CK1α and β-catenin as the top hits required for survival of murine AML leukemic stem cells (LSCs) and AML disease propagation in the sub lethally-irradiated recipient mice. In contrast to AML LSCs, CK1α knockdown did not affect survival of normal HSCs and CK1 inhibition by D4476 resulted in the selective killing of AML LSCs in vitro and their depletion in vivo over normal HSCs. Gene expression profiling of the cells with silenced *Csnk1a* revealed signatures of increased p53 activity and myeloid differentiation, well in line with the known role of CK1α as a regulator of p53 pathway [114,115,116,117,118]. The authors [113] could indeed experimentally confirm that the anti-leukemic effects of CK1α suppression are dependent on p53 - p53 null leukemic cells were resistant to *Csnk1a* silencing. This is a key observation that pointed towards two important facts: (i) the role of CK1α in AML LSCs regulation is complex but likely more connected to p53 pathway than to Wnt/β-catenin pathway, a known driver of AML [119] that is inhibited by CK1α and (ii) the active p53 signaling is a prerequisite for therapeutic targeting of AML/MDS cells by CK1α inhibition. These two aspects are discussed in detail in two paragraphs below.

It has been shown that Wnt/β-catenin signaling pathway is required for self-renewal and survival of AML LSCs [120,121,122] and as such Wnt/β-catenin inhibition represents potential therapeutic strategy for AML [119,123]. CK1α is a well-characterized β-catenin kinase that promotes β-catenin degradation and CK1α suppression or inhibition thus leads to upregulation of β-catenin levels and activation of canonical Wnt pathway [124]. As such one can hypothesize that CK1α inhibition will support AML pathogenesis via positive effects on the Wnt/β-catenin pathway. However, despite the increased levels of β-catenin upon CK1α inhibition/knockdown in AML cells, the pro-apoptotic effects via regulation of p53 seem to dominate.

Mutations in *TP53*, encoding p53, are not common in AML/MDS patients and represent 5–20% de novo cases, however the proportion is increased in the older patients, in AML transformed from MDS, in cases carrying del(5q) or complex karyotypes or in the therapy-induced AML (chemotherapy/irradiation) [105,125]. Overall, the mutations are connected to inferior prognosis and poor response to therapy. Possible CK1α-targeted therapies, which will likely rely on p53 activity, can be hypothetically effective in approx. 80% of AML with wt *TP53* but can be also synergistic with the approaches focused on the restoration of mutant p53 function [126,127].

### 5.2. Targeting CK1α in MDS/AML Patients—Lenalidomide and BTX-A51

The importance of CK1α in del(5q) MDS was uncovered by the studies attempting to understand why lenalidomide is therapeutically active only within this patient subgroup [108,128]. Lenalidomide is a derivative of thalidomide developed to increase immunomodulatory potency and minimize toxicities, FDA-approved for the treatment of transfusion-dependent patients with lower-risk MDS bearing del(5q) mutation with or without additional cytogenetic abnormalities [129]. Lenalidomide has been shown to be able to eradicate del(5q) MDS clones, to inhibit growth of CD34+ cells carrying this aberration [130,131] and to induce cytokinesis defects in the treated del(5q) MDS-L cells [128]. However, its mode of action remained elusive until Kronke et al. [108] described that lenalidomide induces ubiquitination and degradation of CK1α by the E3 ubiquitin ligase CUL4–RBX1–DDB1–CRBN (known as CRL4CRBN) complex. The effects of lenalidomide were shown to be very specific – only three proteins—CRBN, IKZF1 and CK1α—were downregulated; of note *CSNK1A1* mRNA expression was not affected. This mechanism seems to be rather universal as researchers confirmed their findings in multiple human cell lines and in the BM and PB of AML patients treated in vivo by lenalidomide. Importantly, lenalidomide shows therapeutic effect in del(5q) MDS but failed in AML clinical trials [132]. The reasons are unclear but it was proposed that it can be due to synthetic lethality that occurs only in del(5q) MDS with haploinsufficient *CSNK1A1* [109] but fails in AML with normal *CSNK1A1* or in del(5q) AML that often carries *TP53* mutations. In theory, this can be overcome, at least in *TP53* wild type cases, by direct inhibition of CK1α.

Targeting CK1α thus seems to be a promising approach for the treatment of MDS and AML that was, until recently, hampered by the lack of suitable CK1α inhibitors. This issue has been addressed by Minzel, Venkatachalam and colleagues [8], who recently published a preclinical study presenting a family of novel small-molecule kinase inhibitors that can efficiently inhibit CK1α in vivo. One compound from the series, A-51, has recently entered phase I of clinical trial with focus on relapsed or refractory AML/high risk MDS (BTX-A51, NCT04243785). In their preclinical study [8], the authors presented a series of compounds that are potent inhibitors of CK1α and are able to induce p53 activity and upregulate β-catenin as expected. Selected compounds induced apoptosis in AML leukemia progenitor cells in vitro and eradicated them in vivo in both genetically engineered AML mouse models and in the patient-derived xenograft mouse models. Interestingly, the most potent compounds were found to inhibit also transcriptional kinases cyclin-dependent kinase (CDK) 7 and CDK9. The authors propose that the anti-leukemic effect was extended by inhibition of oncogene-driving super-enhancers regulated by CDK7/9 that resulted in downregulation of protooncogenes MYC, MCL1 and MDM2. Despite the high potential of BTX-A51, future work is required to dissect the contribution of inhibition of CDK7/CDK9 but also of other kinases. Based on the publicly available data [8], BTX-A51 efficiently (single/double digit nM Kd) blocks multiple other kinases from CDK, JNK, DYRK and JNK families. Of note, it is also a very potent inhibitor of other CK1 members—including CK1δ and CK1ε—that has been shown to regulate MYC and MDM2 [4,57].

## 6. Summary and Future Directions

CK1 kinases are critically involved in multiple cellular processes that appear to be valuable therapeutic targets in leukemia. No specific CK1 inhibitors, however, have reached the clinical stage yet, although two compounds that among others target also CK1 are currently in the clinical trials. Umbralisib, a dual PI3Kδ and CK1ε inhibitor, is currently under review for approval by the FDA for indolent non-Hodgkin lymphoma. Second, BTX-A51 that targets CK1α, CK1δ, CK1ε, CK1γ and also multiple other kinases including CDK7 and CDK9 [8] is in a phase I AML clinical trial.

Both the mechanism of action and the CK1 isoforms targeted by these two drugs are different. In CLL, the CK1δ/ε inhibition slows down disease progression in preclinical models, likely via inhibition of the non-canonical Wnt pathway and by disruption of the microenvironmental interactions [30]. Umbralisib brings benefits in comparison to other PI3K inhibitors used in CLL [133]. On the other side BTX-A51 is believed to act primarily via inhibition of CK1α and subsequent activation of p53-dependent cell death [8]. Currently it is difficult to judge how the inhibition of other kinases contributes to the possible therapeutic benefit but also to the unwanted side effects. This status, however, encourages the development of more specific CK1 inhibitors—with improved CK1 isoform specificity and a limited number of off-targets from other kinase families. Novel specific CK1ε inhibitors could play an important role in the therapeutic strategy for refractory or relapsed patients with CLL and NHL, whereas CK1α specific inhibitors can be useful for MDS and some subtypes of AML.

Several pan-CK1 or CK1δ/ε dual inhibitors appear to be well tolerated in mouse [8,30,134] and umbralisib data suggests that CK1ε inhibition is also well tolerated in human patients [96,133]. However, CK1 kinases bear an essential role in many homeostatic processes and the redundancy of individual CK1 isoforms (mainly CK1δ and CK1ε) is common. For example in the intestinal stem cell compartment only ablation of both CK1δ and CK1ε depletes stem cells and disrupts intestinal architecture [135]. It is thus probable that pan-CK1 or CK1δ/ε dual inhibitors will have higher on target toxicity than the CK1 isoform specific compounds.

It remains to be studied which molecular and cellular mechanisms dominantly contribute to the effects of CK1 inhibition in preclinical models and in clinical trials. Their actions can be cell intrinsic as well as more complex including interaction with the microenvironment and cell-cell communication. CK1 is a critical regulator of multiple signaling pathways including Wnt, Hedgehog and Yap/Taz (for reviews see [4,14]). CK1 inhibition can also have a more general immunomodulatory effect. The non-canonical Wnt signaling pathway, which is dependent on CK1, is known to mediate inflammatory response and WNT-5A has been implicated in many inflammatory conditions such as rheumatoid arthritis [136], osteoarthritis [137] and psoriasis vulgaris [138] or recently has been described as a reliable biomarker for monitoring of pathological progression in SARS-CoV-2 patients [139]. In line with these observations, CK1ε inhibition was able to block cartilage destruction in the mouse osteoarthritis model [137]. In addition, CK1 inhibition can modulate T cells—CK1δ was shown to be essential for the formation of the immunological synapse [140] and CK1ε inhibition by umbralisib and PF-4800567 could protect Tregs and prevent immune system mediated toxicities in CLL patients treated with PI3K inhibitors [96,133].

Last but not least, the potential use of CK1 inhibitors is not limited to the hematological malignancies discussed in detail in the previous chapters. There is growing evidence for the critical role of CK1α in multiple myeloma (MM), which is a cancer of plasma cells. CK1α has been proposed, based on the shRNA-mediated gene knockdown and treatments with D4476 inhibitor as a critical regulator of MM that promotes plasma cell proliferation and survival [141]. Depletion/inhibition of CK1α is cytotoxic to MM cells but is dispensable for normal B cells and, importantly, synergizes with the clinically used proteasome inhibitors (bortezomib) and immune modulators (lenalidomide) [142]. CK1α inhibition in MM was proposed to act via several mechanisms including the inhibition of the pro-survival autophagy [143,144], regulation of MYC [141] or by unleashing p53-driven apoptosis [142].

In summary, individual CK1 kinases appear to be essential players in multiple hematological malignancies. They act by various mechanisms and provide therapeutic opportunities for several poorly manageable clinical conditions. However, further preclinical research and clinical trials are needed to assess the full potential of CK1 inhibition in hematologic cancers.

## Figures and Tables

**Figure 1 ijms-21-09026-f001:**
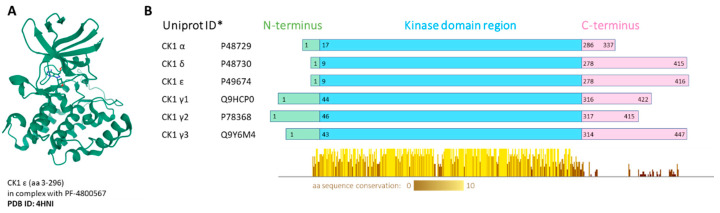
(**A**) Visualization of human CK1ε crystal structure in complex with PF-4800567 bound in the adenosine triphosphate (ATP)-binding pocket. The structure screenshot is based on the PDB 4HNI [10] entry from the RCSB PDB (rcsb.org) [11]. (**B**) Conserved regions visualization. Multiple alignment was produced by ClustalOmega, the conserved regions present in the kinase domain were visualized by Jalview 2.11.0. * Canonical variant; aa—amino acid.

**Figure 2 ijms-21-09026-f002:**
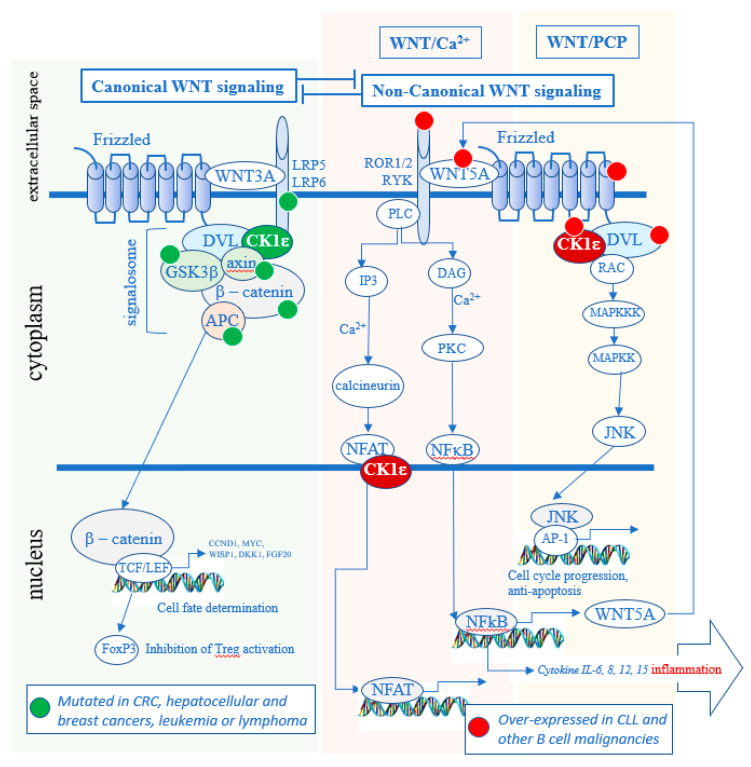
Role of CK1δ/ε in the Wnt signaling pathways. In the canonical pathway, it maintains the signalosome active and blocks the degradation of β-catenin. In the non-canonical pathway, CK1δ/ε kinase activity leads to the activation of the Wnt/Ca2+ and Wnt/PCP pathways. Many oncogenic mutations in the canonical pathway are found in solid tumors (green dots) [23,24,25,26,27], whereas many over-expressions of oncogenes are present in hematologic cancers (red dots) [28,29,30,31,32]. CRC—colorectal cancer.

**Table 1 ijms-21-09026-t001:** Reported in vitro kinase assay IC_50_ values and the most common off-targets of the CK1 kinase inhibitors used in the studies discussed in this review.

IC_50_ (nM)/ % Kinase Activity	CK1ε	CK1δ	CK1α	CK1γ1/2/3	Other Targets	Source Publication	Stage
PF-4800567	32	711	-	No	EGFR	[54]	Preclinical
PF-670462	7.7	14	-	No	EGFR, p38α,	[54,56]	Preclinical
D4476	270	300	37% at 0.5 μM *	No	ALK5	[48,53,56]	Preclinical
BTX-A51	4.4	1.8	5.3	20/0.5/6	CDK7/9	[8]	Clinical, Phase I
Umbralisib	40% at 1 μM *	105% at 1 μM *	111% at 1 μM *	96–104% at 1 μM *	PI3Kδ	[57]	Clinical, Phase III

The kinase domains of CK1ε and δ are 98% identical, therefore most CK1ε inhibitors also block CK1δ (and vice versa) with the exception of PF-4800567 (22-fold selectivity for CK1ε) and umbralisib (no detected CK1δ inhibition). Hyphen—data not provided in the original publication; No—no detected kinase inhibition in the assay; * IC_50_ not available, data represent % of the kinase activity at the indicated inhibitor concentration.

**Table 2 ijms-21-09026-t002:** Duvelisib and idelalisib are two PI3K inhibitors approved for therapy of B cell malignancies.

*Kd (nM)*	*PI3Kδ*	*PI3Kγ*	*PI3Kα*	*PI3Kβ*	*CK1ε*
***Duvelisib***	0.047	0.21	40	0.89	>30000
***Idelalisib***	1.2	9.1	600	19	>30000
***Umbralisib***	6.2	1400	>10000	>10000	180

Umbralisib filling for FDA approval in FL and MZL lymphoma is ongoing. Umbralisib is the only PI3K inhibitor described so far to also inhibit CK1ε [93].

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
