# Peer review of "Targeting Casein Kinase 1 (CK1) in Hematological Cancers"

_ijms, 2020, doi:10.3390/ijms21239026_

Round 1
Reviewer 1 Report
In the present review Janovska et al provide an updated overview of the role of the casein kinase (CK1) family of enzymes in hematological malignancies, with special focus on the development of CK1 inhibitors with therapeutic potential.
Overall, I find the review well written and very informative, as it includes a lot of the recent work on CK1 inhibitor research. I do not have any major concerns or corrections.
As a minor comment, I would suggest a rephrasing of the sentence included in lines 122-124 (page 4), since the "not only" included in parenthesis is confusing for the reader. Hematological malignancies are indeed cancers, so the sentence does not really make sense as it stands. I assume the authors want to stress the potential importance of CK1 inhibitors in non-hematological malignancies (as stated in the title of the section).
Reviewer 2 Report
Diligent and comprehensive review article targeting all relevant aspects of the role of CK1 enzymes for different lymphatic and myeloid malignancies. Preclinical and clinical studies evaluating different CK1 inhibitors are described. No critizisms from my side.
